# PROFILE-AWARE MANEUVERING: A DYNAMIC MULTI-AGENT SYSTEM TO ROBUST AGENTIC PROBLEM SOLVING

## ABSTRACT

The rapid advancement of large language models (LLMs) has empowered intelligent agents to leverage external tools for solving complex problems, yet this reliance introduces new challenges as extended contexts and noisy tool outputs undermine system reliability. We argue that building robust agents requires the rigor of control engineering, rather than relying on empirical prompt engineering. Drawing inspiration from predictive control in vessel maneuvering, we reframe agent design as a formal control systems problem. We first establish a baseline Multi-Agent System (MAS) where a Guard Agent acts as a simple reactive feedback controller, correcting a primary Execution Agent's errors after they occur. However, this reactive approach is fundamentally limited. Our core contribution, termed Profile-Aware Maneuvering, elevates this to a predictive control architecture. Through an automated offline 'System Identification' process, we generate an explicit, text-based 'performance fingerprint' modeling the Execution Agent's characteristic failure modes. Armed with this fingerprint, the Guard Agent evolves from a reactive critic into a predictive controller. It implements a feed-forward strategy to preemptively counteract errors before they derail the reasoning process. Experiments across a spectrum of benchmarks, including GAIA, HLE, and GPQA Diamond, validate our approach. The final Profile-Aware MAS demonstrates the hallmarks of a well-controlled system: it dramatically reduces performance variance while simultaneously boosting accuracy, and it minimizes the gap between its potential and single-pass performance. This superior performance and stability culminated in our system achieving a score of over 81 on the GAIA leaderboard. Our findings advocate for a paradigm shift: from the empirical art of prompt engineering to the principled science of control theory for designing predictable and trustworthy intelligent agents.

## 1 INTRODUCTION

The extraordinary progress of Large Language Models (LLMs) in both capability and scale (Achiam et al., 2023; Touvron et al., 2023; Team et al., 2023; The Google DeepMind Team, 2024; Anthropic, 2025) has sparked widespread curiosity about the upper bounds of artificial intelligence. As practitioners push these frontiers, it has become clear that augmenting foundational models with external tools not only expands their problem-solving abilities beyond intrinsic knowledge but also enables the tackling of complex real-world challenges (Kapoor et al., 2024; Huang & Yang, 2025; Krishnan, 2025; Shao et al., 2025). A vivid illustration is the recent IMO competition, where state-of-the-art LLMs struggled in isolation, whereas agent-based systems built upon them solved most tasks (Huang & Yang, 2025). This paradigm shift suggests the next frontier lies not just in the raw power of individual models, but in the principled architecture of their collaboration.

This insight has fueled the rapid growth of multi-agent frameworks, yet amid the excitement, a central challenge has emerged: system stability. Empirical results show that agent robustness hinges on the foundational model's reliability, the nature of integrated tools, and the design of agent orchestration (Coletta et al., 2024; Li et al., 2025; Shojaee et al., 2025). For instance, while promising, common paradigms like the "solver–reviewer" structure often rely on rigid, turn-based dialogues. This can lead to bloated contexts and, more critically, an inability to intervene at the precise moment

of failure. This exposes a foundational gap: the absence of a control strategy for building agents that are not just collaborative, but also predictably consistent, resilient, and adaptive.

Drawing inspiration from control theory, specifically vessel maneuvering—where a ship's autopilot uses dynamic adjustments, not static settings, to maintain its course (Xie et al., 2020)—we argue that intelligent agents require a similar principle of dynamic maneuvering. Instead of relying on fixed supervision, agents should adaptively decide when and how to intervene based on the evolving context. To realize this, we constructed a dynamic Multi-Agent System (MAS) within our open-source multi-agent framework. This architecture establishes a reactive feedback loop: the Guard Agent corrects the system's trajectory based on observed errors. This is a crucial first step, but one that can only fix deviations after they have already occurred.

To transcend this reactive limitation and achieve proactive guidance, we address the system's critical blindness to its partner's habitual failure modes. This motivates our core contribution, a novel form of Context-Level Reinforcement inspired by System Identification (Xu & Soares, 2013; Xue et al., 2021; Alexandersson et al., 2024). Instead of tuning internal model weights via implicit rewards, we reinforce the agent's reasoning process at the system level. In a preparatory offline stage, we systematically benchmark the Execution Agent to generate a 'performance fingerprint'—an explicit, human-readable policy detailing its characteristic errors, such as a tendency to hallucinate code. During online execution, this fingerprint is injected directly into the Guard Agent's context, empowering it to provide profile-aware, preemptive guidance and transforming it into an expert on its specific partner.

This shift towards proactive control fundamentally distinguishes our work from prevailing agent improvement paradigms rooted in cognitive learning theories. Frameworks such as ReAct (Yao et al., 2023), which synergizes reasoning and acting in real-time, or Reflexion (Shinn et al., 2023) and Expel (Zhao et al., 2024), which rely on post-hoc verbal reflection and the retrieval of past experiences, are inherently reactive or adaptive. They aim to correct errors after they manifest or adapt to new tasks based on prior successes. In stark contrast, our methodology is explicitly predictive. The offline System Identification process does not learn from task-specific successes, but rather models the agent's intrinsic, task-agnostic failure modes. This enables a feed-forward control strategy that preempts errors before they occur, shifting the objective from building agents that learn from failure to engineering systems that are designed to avoid it.

Rigorous testing on the GAIA benchmark (Mialon et al., 2023) provides empirical proof for our control-theoretic thesis. The final Profile-Aware MAS, orchestrated with a predictive control architecture, not only surpasses simpler systems but exhibits the hallmarks of a well-controlled system: superior stability (Tables 1) and reliability. This culminated in a score over 81 on the GAIA test leaderboard. This paper will demonstrate that by reframing agent design as a control systems problem, we can engineer intelligent systems that are not only more capable, but fundamentally more predictable and trustworthy.

## 2 METHOD

Our methodology is grounded in the principles of Engineering Cybernetics and Control Theory, which provide a robust framework for designing and analyzing complex, dynamic systems. We first establish our theoretical foundation by drawing a parallel with the well-understood problem of marine vessel maneuvering. We then formalize the LLM-based agent as a controllable system. Finally, we detail our progressively sophisticated control architectures, interpreting them through the rigorous lens of control theory.

### 2.1 THEORETICAL FOUNDATION: MANEUVERING COMPLEX SYSTEMS

The core challenge in controlling any complex system, from a supertanker to an intelligent agent, is to ensure its behavior converges to a desired trajectory despite internal instabilities and external disturbances. In Engineering Cybernetics, this is achieved by first understanding the system's intrinsic dynamics and then designing an appropriate control strategy (Tsien & Qian, 1954).

A perfect, tangible illustration of this principle is found in marine vessel navigation (Xie et al., 2020; Alexandersson et al., 2024). The motion of a ship is governed by a set of linearized equations.

Critically, these equations contain hydrodynamic coefficients ($X_{\dot{u}}$, $Y_v$, etc.) that are unique to each vessel. These unknown parameters must be experimentally determined through a process called System Identification, where standardized tests (like the zig-zag maneuver shown in Figure 1) are used to create a precise mathematical 'fingerprint' of the ship's behavior. Only with this fingerprint can an effective control system (e.g., an autopilot) be designed to actively counteract disturbances and guide the ship along a desired path.

We posit that this two-stages: first, identify the system's unique behavioral fingerprint, then design a targeted control architecture and directly apply to engineering reliable LLM-based agents.

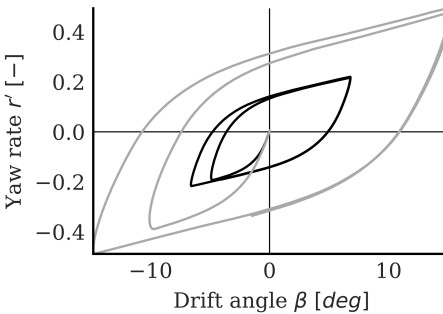

Figure 1: The zig-zag test is a standard procedure in System Identification for marine vessels, to reveal the ship's unique maneuvering characteristics (its 'fingerprint') (Alexandersson et al., 2024).

## 2.2 MODELING THE AGENT AS A CONTROLLABLE SYSTEM

To apply the rigorous discipline of control theory, we first formalize the agent's problem-solving process as a controllable system. The foundational LLM, a static function $y = f_\theta(x)$ with immutable parameters $\theta$, serves as the core of our system. Our entire methodology is an exercise in applied control engineering: how to strategically design the input context, $x$, to steer the output, $y$, towards a correct and stable solution.

Before detailing our architectures, we establish a mapping between control theory variables and their concrete instantiations within our agent framework:

**Plant** ($P$): The Execution Agent ('E'), the core process to be controlled. Its behavior is dictated by $f_\theta$.

**Controlled Variable** ($y$): The Execution Agent's output, $y_E$, the variable we want to regulate.

**Setpoint** ($r$): The implicit, desired 'correct' reasoning path. This is the target trajectory.

**Error** ($e$): The deviation of the agent's output from the correct path, $e = r - y_E$. This is often detected implicitly as logical fallacies or factual inaccuracies.

**Disturbance** ($d$): Internal factors (e.g., hallucinations, logical fallacies) and external factors (e.g., noisy tool outputs) that cause $y_E$ to deviate from $r$.

**Controller** ($C$): The Guard Agent ('G'), which observes the system and computes a corrective action. Its behavior is also dictated by $f_\theta$.

**Control Signal** ($u$): The critique or guidance $y_G$ generated by the Guard Agent. This is the action applied to steer the plant.

## 2.3 HIERARCHICAL CONTROL ARCHITECTURES FOR AGENT MANEUVERING

We designed and evaluated a hierarchy of control architectures, each representing a more sophisticated control strategy, as depicted in Figure 2.

### 2.3.1 THE UNCONTROLLED SYSTEM (SINGLE AGENT SYSTEM)

In the baseline case, the agent operates in an 'open loop' without a controller. The control signal is effectively zero ($u = 0$). At each step $t$, the agent's context $x_t$ is formed by concatenating the task $x_{\text{task}}$, its prior reasoning $\mathcal{T}_t = \{y_{E,0}, ..., y_{E,t-1}\}$, and new tool information $\text{Info}_t$. The agent's next action is then:

$$y_{E,t} = f_\theta(x_{E,t}) \quad \text{where} \quad x_{E,t} = x_{\text{task}} \oplus \mathcal{T}_t \oplus \text{Info}_t$$

This system relies solely on the intrinsic capabilities of the plant ($f_\theta$) and is highly susceptible to disturbances, analogous to a ship drifting without rudder control.

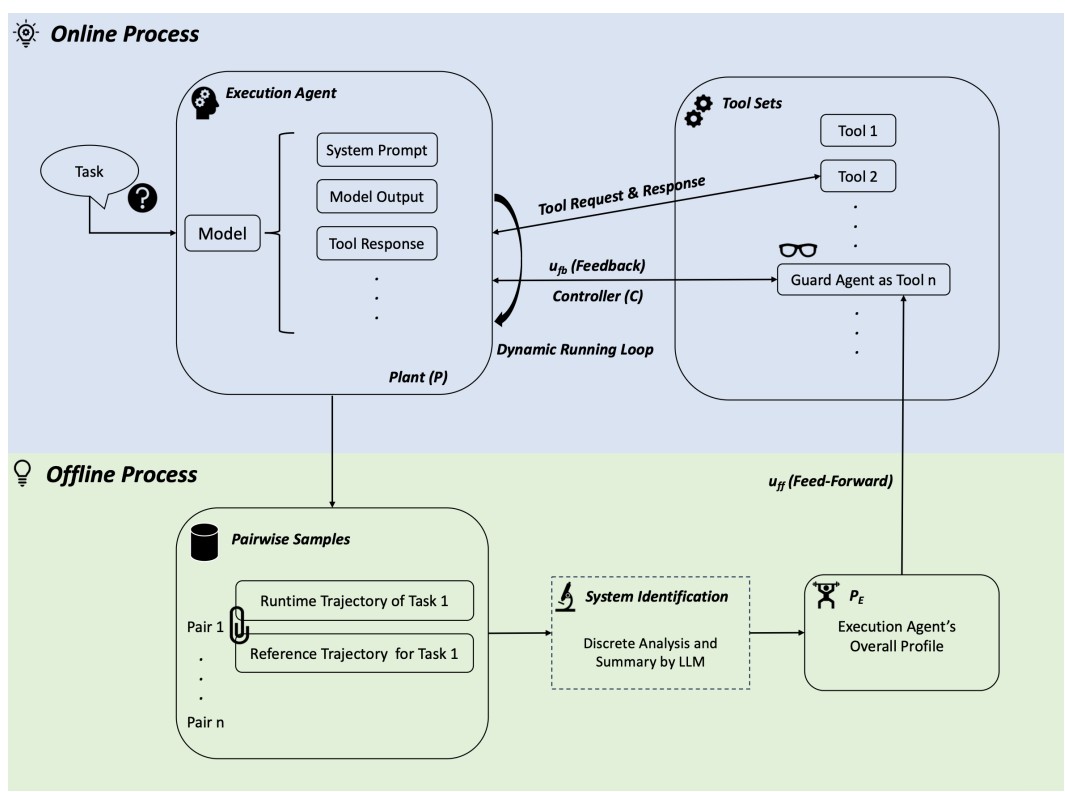

Figure 2: Our hierarchical control architectures, built on our framework. This figure illustrates the components for the Single Agent System (uncontrolled), the Naive MAS (feedback control), and the Profile-Aware MAS (composite feed-forward-feedback control), which leverages a fingerprint from an offline System Identification process.

### 2.3.2 REACTIVE FEEDBACK CONTROL (THE NAIVE MAS)

Our first control strategy implements a classic negative feedback loop. A Guard Agent ('G') is introduced to act as the controller. In standard control theory, the controller's action is defined by the feedback law $u = C(e)$. In our framework, this is realized as follows:

- The Guard Agent observes the plant's output trajectory, $\mathcal{T}_{E,t+1}$.
- By analyzing this trajectory for logical flaws, it implicitly computes the error $e$.
- It then generates a critique, which serves as the control signal $u$.

This process is formally described as:

$$u_t = y_{G,t} \quad \text{where} \quad y_{G,t} = f_\theta(\mathcal{T}_{E,t+1})$$

This control signal $u_t$ s fed back into the agent's context, closing the loop:

$$x_{E,t+1} = x_{E,t} \oplus u_t$$

This architecture mirrors a standard feedback controller, which is effective for stabilization but is fundamentally reactive, it can only correct an error after it has already occurred and been detected.

### 2.3.3 PREDICTIVE COMPOSITE CONTROL (THE PROFILE-AWARE MAS)

The pinnacle of our approach is a sophisticated composite feed-forward-feedback control system. This strategy first requires creating a model of the plant's predictable behaviors via System Identification (Xu & Soares, 2013; Xue et al., 2021; Alexandersson et al., 2024).

**System Identification**: We subject the Execution Agent ('P') to the validation sets of the specific benchmark, analyzing its logs along with the reference step and answer, forming the sample pair, to identify characteristic failure modes in response to certain task types (disturbances, 'd'). This analysis is synthesized into a performance fingerprint, $\mathcal{P}_E$, a structured textual policy that is human-readable and directly inserted into the Guard Agent's context.

**Online Serving with the Guard Agent**: In the online phase, this fingerprint enables a composite control law, $u = u_{fb} + u_{ff}$, where $u_{fb}$ is the reactive feedback signal and $u_{ff}$ is a proactive feed-forward signal. Our non-linear controller $f_\theta$ generates this composite signal in a single inference step:

$$u'_t = y'_{G,t} \quad \text{where} \quad y'_{G,t} = f_\theta(\underbrace{\mathcal{T}_{E,t+1}}_{\text{for } u_{fb}} \oplus \underbrace{\mathcal{P}_E}_{\text{for } u_{ff}})$$

Here, the Guard Agent processes $\mathcal{T}_{E,t+1}$ to generate the reactive feedback component ($u_{fb}$) for any observed error, while simultaneously processing $\mathcal{P}_E$ to anticipate errors characteristic of the Execution Agent and generate a pre-emptive feed-forward component ($u_{ff}$). This proactive guidance is the hallmark of a truly robust control system.

### 2.3.4 THEORETICAL JUSTIFICATION OF COMPOSITE CONTROL

We now provide a formal justification from control theory to prove why a composite system is theoretically superior for handling predictable disturbances. Using Laplace transforms, where functions of time $t$ become functions of a complex variable $s$, we analyze the system's response to a disturbance $D(s)$.

**Limitation of Pure Feedback Control (Naive MAS):**  In a standard feedback system, the relationships between the components are defined as follows: the system output $Y(s)$ is the sum of the plant's action and the disturbance; the control signal $U(s)$ is the controller's action on the error; and the error $E(s)$ is the difference between the setpoint $R(s)$ and the output.

$$Y(s) = P(s)U(s) + D(s) \tag{1}$$
$$U(s) = C(s)E(s) \tag{2}$$
$$E(s) = R(s) - Y(s) \tag{3}$$

By substituting (3) into (2), and then into (1), we can solve for the closed-loop output $Y(s)$:

$$Y(s) = \frac{P(s)C(s)}{1 + P(s)C(s)}R(s) + \frac{1}{1 + P(s)C(s)}D(s)$$

From this equation, it is evident that the system can only suppress the disturbance $D(s)$ by increasing the controller gain to make the term $1 + P(s)C(s)$ large. It cannot eliminate the disturbance's effect entirely, as the controller only acts *after* $D(s)$ has already corrupted the output $Y(s)$. This is the mathematical definition of a reactive system.

**Superiority of Composite Control (Profile-Aware MAS):**  With a feed-forward controller $C_{ff}$ added, the total control signal becomes a composite of the feedback signal $U_{fb}(s)$ and the feed-forward signal $U_{ff}(s)$.

$$U(s) = U_{fb}(s) + U_{ff}(s) \tag{4}$$
$$U_{fb}(s) = C_{fb}(s)E(s) = C_{fb}(s)(R(s) - Y(s)) \tag{5}$$
$$U_{ff}(s) = C_{ff}(s)D(s) \tag{6}$$

Substituting this composite control law into the plant equation (1):

$$Y(s) = \frac{P(s)C_{fb}(s)}{1 + P(s)C_{fb}(s)}R(s) + \frac{1 + P(s)C_{ff}(s)}{1 + P(s)C_{fb}(s)}D(s)$$

Perfect rejection of the disturbance is theoretically possible if we can design a feed-forward controller $C_{ff}(s)$ such that the numerator of the disturbance term becomes zero:

$$\text{If } C_{ff}(s) = -\frac{1}{P(s)}, \quad \text{then the term } 1 + P(s)C_{ff}(s) = 0$$

This would cancel the disturbance preemptively, before it affects the output.

The performance fingerprint $\mathcal{P}_E$ serves as our empirically-derived model of the plant $P(s)$. The ideal feed-forward controller, $C_{ff}(s) = -1/P(s)$, requires a controller that acts as the negative inverse of the plant. In our system, this 'negative inverse' is not achieved via a literal sign but through the learned function of the Guard Agent. The Guard Agent's role is defined as a critiquer and corrector. Its fundamental objective is to generate an output, a critique, whose semantic effect is to invert and cancel the predicted error of the Execution Agent. This core corrective purpose of the Guard Agent embodies the negative sign required by control theory, ensuring the feed-forward action suppresses, rather than amplifies, disturbances. This formal analysis proves that our profile-aware architecture is theoretically superior for achieving robust, predictive, and stable agent maneuvering.

## 3 EXPERIMENTS SETTINGS

### 3.1 GAIA PROBLEM SET

Our experiments utilize 109 questions (the specific task IDs will be released later on GitHub) from the GAIA test set (Mialon et al., 2023), comprising 56 Level 1 (L1) and 53 Level 2 (L2) questions. These questions cover a range of tasks, including office-related activities such as working with Excel, Word, PowerPoint, text files, code, and download tools, as well as search-related operations involving resources like Google Search and Wikipedia. To ensure a fair comparison of different agent construction methodologies, the experimental setup minimizes external influences such as browser instability, maintaining a controlled environment throughout. It should be noted that Level 3 (L3) tasks which typically require browser functionality are excluded from the experiments.

### 3.2 HLE AND GPQA DIAMOND SET

To verify the broad applicability of our approach, we conducted additional experiments on two distinct benchmarks: Humanity's Last Exam (HLE) and GPQA Diamond. These datasets were chosen to test our method's performance under varying levels of difficulty.

Humanity's Last Exam (HLE) is a challenging benchmark from the Center for AI Safety and Scale AI, featuring 2,500 questions across a wide range of subjects. We randomly selected a subset of 100 questions (47 Math, 15 Biology/Medicine, 11 Computer Science/AI, 10 Physics, and 17 from other fields, task IDs will be released later). From this subset, we first ran the SAS to establish a baseline. Then, 10 random question-answer logs were chosen for the offline System Identification process to generate the Execution Agent's performance fingerprint. The remaining 90 questions served as the test set to evaluate and compare the performance of the SAS, the MAS, and our Profile-Aware MAS.

GPQA Diamond is a high-quality subset of the GPQA benchmark, comprising 198 expert-level multiple-choice questions in biology, physics, and chemistry. Following a similar protocol, we randomly selected 12 questions for System Identification on the SAS. The performance of the SAS and our Profile-Aware MAS was then compared on the remaining 186 questions.

### 3.3 EXPERIMENTAL VERSION DESIGN

We compare four distinct methodologies in our experiments. First, the Base approach involves direct question-answering by a single Gemini 2.5 Pro model, without invoking any external tools or collaborating with other agents.

Second, the Single Agent System (SAS) pairs the same foundational model (Gemini 2.5 Pro) with a detailed system prompt and various MCP tools. Here, the model autonomously decides, based on the question and context, whether to use external tools or to answer independently.

Third, our Multi-Agent System (MAS) extends the SAS setup by introducing the dynamic supervision mechanism. This is achieved by building a Guard Agent as an additional candidate tool,

which the Execution Agent can engage for real-time logical verification during the problem-solving process. In this configuration, the Guard Agent provides 'naive' supervision, as it has no prior knowledge of the Execution Agent's specific tendencies.

Finally, our Profile-Aware MAS enhances this architecture with our core contribution inspired by System Identification. It builds directly upon the MAS but equips the Guard Agent with a 'performance fingerprint' of its partner. This fingerprint is generated in a preparatory offline stage, where the Execution Agent's behavior is systematically benchmarked on a separate dataset to identify its characteristic failure modes. During the online evaluation, the Guard Agent leverages this fingerprint to provide profile-aware supervision, making targeted interventions based on its partner's known weaknesses rather than merely reacting to immediate logical inconsistencies.

## 3.4 RUNNING SETTINGS

Each experiment consists of three independent runs across the test tasks for every version, all utilizing the Gemini 2.5 Pro model with a temperature setting of 0.1. If a task yields an answer in an invalid format, it is repeated until a valid response is obtained. For each run, we report the Pass@1 accuracy, and for each version, we also report the aggregated Pass@3 accuracy across all runs.

## 4 EXPERIMENTAL RESULTS

Our empirical evaluation validates the control-theoretic approach across three diverse benchmarks: GAIA, a benchmark for general-purpose AI agent capabilities; Humanity's Last Exam (HLE), a high-difficulty academic question-answering dataset; and GPQA Diamond, a high-baseline expert-level multiple-choice dataset. The results, summarized in Tables 1 and 2, demonstrate a clear, progressive improvement across raw accuracy, system stability, and reasoning reliability.

Table 1: GAIA Benchmark Results: Detailed performance summary from the baseline LLM to the Profile-Aware MAS (PA-MAS), with relative comparisons ('vs' columns denote percentage change).

| Metric | LLM | SAS | vs LLM (%) | MAS | vs SAS (%) | PA-MAS | vs MAS (%) |
|---|---|---|---|---|---|---|---|
| Round 1 P@1 | 32.11% | 56.88% | | 70.64% | | 72.48% | |
| Round 2 P@1 | 30.28% | 63.30% | | 64.22% | | 70.64% | |
| Round 3 P@1 | 32.11% | 64.22% | | 66.06% | | 69.72% | |
| Pass@3 | 38.53% | 80.73% | +109.53 | 82.57% | +2.28 | **84.40%** | +2.22 |
| Pass@1_avg | 31.50% | 61.47% | +95.14 | 66.97% | +8.95 | **70.95%** | +5.93 |
| Pass@1_std | 0.0086 | 0.0327 | +279.07 | 0.0270 | -17.18 | **0.0115** | -57.41 |
| P@3-P@1_avg | 0.0703 | 0.1926 | +173.97 | 0.1560 | -19.02 | **0.1345** | -13.75 |

Table 2: Detailed comparative performance across the high-difficulty HLE and high-baseline GPQA Diamond benchmarks. This comprehensive table includes raw single-run data and aggregated metrics with relative comparisons ('vs' columns denote percentage change).

| | HLE Benchmark | | | | | GPQA Diamond Benchmark | | |
|---|---|---|---|---|---|---|---|---|
| Metric | SAS | MAS | vs SAS (%) | PA-MAS | vs MAS (%) | SAS | PA-MAS | vs SAS (%) |
| Round 1 P@1 | 15.56% | 25.56% | | 20.00% | | 86.02% | 88.17% | |
| Round 2 P@1 | 18.89% | 15.56% | | 25.56% | | 80.11% | 85.48% | |
| Round 3 P@1 | 16.67% | 12.22% | | 18.89% | | 83.87% | 86.56% | |
| Pass@3 | 27.78% | 33.33% | +19.98 | **35.56%** | +6.69 | 90.91% | **92.42%** | +1.66 |
| Pass@1_avg | 17.04% | 17.78% | +4.34 | **21.48%** | +20.81 | 83.33% | **86.74%** | +4.09 |
| Pass@1_std | **0.0138** | 0.0567 | +310.87 | 0.0292 | -48.50 | 0.0244 | **0.0111** | -54.75 |
| P@3-P@1_avg | **0.1074** | 0.1555 | +44.79 | 0.1408 | -9.45 | 0.0758 | **0.0568** | -25.07 |

**Accuracy Progression: A Consistent and Scalable Ascent.** On the GAIA benchmark, introducing tools (SAS), reactive feedback (MAS), and predictive control (Profile-Aware MAS) elevates the Pass@1_avg from 31.50% to 61.47%, 66.97%, and finally to a peak of 70.95%. This positive trend is amplified in high-difficulty scenarios; on the challenging HLE benchmark, the Profile-Aware MAS

delivered a substantial 20.81% relative Pass@1 gain over the MAS, demonstrating that predictive guidance is critical when the agent is prone to error. Conversely, even in the high-baseline GPQA scenario where SAS performance is already strong (83.33%), our Profile-Aware MAS still provided a consistent 4.09% relative accuracy lift. This confirms that each layer of our control architecture contributes distinct value, and the benefits scale with task complexity.

**System Stability: Taming Variance Across the Difficulty Spectrum.** On GAIA, our control strategies progressively tame the instability introduced by tools: the reactive MAS reduces variance by 17.18%, and the predictive Profile-Aware MAS slashes it by a further 57.41%. This powerful stabilizing effect is confirmed on the GPQA benchmark, where the Profile-Aware MAS again cut variance by over 54%. The HLE results reveal a more nuanced dynamic; while reactive control initially increased variance, our predictive method reverses this trend, cutting variance by 48.50%. This consistent stabilization across benchmarks is the empirical manifestation of our control-theoretic approach. As predicted by our Method Section, the feed-forward component ($u_{ff}$) derived from the performance fingerprint preemptively cancels predictable disturbances ($D(s)$), creating a well maneuvered system with superior predictability, a feat reactive feedback alone cannot achieve.

**Reasoning Reliability: Closing the Gap Between Potential and Performance.** The Pass@3 - Pass@1_avg metric quantifies reasoning reliability by measuring the gap between a system's potential (Pass@3) and its typical single-pass performance. On GAIA, the SAS exhibits a large 'regret' gap of 19.26%, which is progressively narrowed by the MAS (15.60%) and the Profile-Aware MAS (13.45%). The GPQA Diamond results mirror this trend, with the Profile-Aware MAS tightening the gap by a significant 25.07%. This demonstrates that our method makes the agent's reasoning more deterministic and less susceptible to stochastic failures. As with the variance metric, the HLE results show an initial widening of this gap for the MAS before the Profile-Aware MAS begins to narrow it. This supports our hypothesis that mastering a complex domain first involves a phase of exploratory instability, which is subsequently stabilized by predictive control.

This trifecta of improvements: higher accuracy, lower variance, and a smaller potential-performance gap provide conclusive evidence that applying a profile-aware, composite control strategy is a superior paradigm for engineering dependable and high-performing intelligent agents.

## 5 ANALYSIS

### 5.1 DYNAMIC MANEUVERING: CONTEXT OPTIMIZATION AND LOGICAL CONVERGENCE

While integrating tools boosts accuracy, the resulting increase in context length introduces significant solution instability. Our GAIA experiments quantify this trade-off: the Pass@1 standard deviation of our tool-augmented Single Agent System (SAS) rises sharply compared to the baseline. To address this, we introduce a dynamic maneuvering mechanism where the Execution Agent can invoke an on-demand Guard Agent upon reaching a logical impasse. This 'second pair of eyes' re-optimizes the context by identifying fallacies, generating a precise prompt to reorient the primary agent's focus, and breaking it out of logical dead ends. Further details are provided in Appendix 3.

This intervention proved highly effective. On the GAIA benchmark, the introduction of the Guard Agent reduced the Pass@1 standard deviation by 17.18%, while enhancing the Pass@1_avg by 8.95%, relative to the SAS. This substantial gain in stability and logical consistency validates our two-agent approach and serves as a foundational step toward more advanced predictive control.

### 5.2 BEYOND PARAMETER TUNING: SYSTEM-LEVEL REINFORCEMENT THROUGH EXPLICIT POLICY

Prevailing methods like Reinforcement Learning (RL) steer agent behavior by fine-tuning millions of opaque internal parameters, an implicit policy distributed within a black-box model (Cheng et al., 2025; Wang et al., 2025; Yan et al., 2025). This makes the resulting policy difficult to interpret, audit, or directly control. In contrast, our work introduces Context-Level Reinforcement, a different philosophy that reinforces the agent's reasoning process at the system level. Instead of back-propagating a scalar reward, we synthesize offline analysis into an explicit, human-readable textual policy, a 'performance fingerprint', which is injected directly into the Guard Agent's context to guide its reasoning path in real-time.

This distinction is analogous to the difference between subconscious skill acquisition and conscious, expert execution. Whereas RL is akin to developing an athlete's muscle memory through repetitive trial and error, our approach is like equipping a pilot with a dynamic, pre-flight checklist tailored to their known habits. It augments the agent's instinct (the internal model) with articulated, explicit best practices (the performance fingerprint). Further details are provided in Appendix 5 and 6.

The empirical evidence validates this philosophy. Our Profile-Aware MAS consistently achieved simultaneous gains in performance and stability: On GAIA, Pass@1 accuracy rose by 5.93% while standard deviation fell by a remarkable 57.41%. On GPQA, this pattern was replicated, with a 4.09% accuracy gain and a 54.75% reduction in variance. On the high-difficulty HLE benchmark, the effect was even more pronounced, with accuracy soaring by 20.81% while reversing the trend towards instability.

These results demonstrate that reinforcing an agent with transparent, external knowledge is a highly effective strategy for achieving more optimal and stable behavior.

# 6 FUTURE WORK

Our work lays the groundwork for several future research directions, each progressively expanding the scope of the profile-aware paradigm:

**Tool-Augmented Guardian for Fact-Checking:** Empowering the Guard Agent with external tools (e.g., search, code interpreters) would advance it from a logic referee to an active fact-checker. By independently verifying the Execution Agent's output against external realities, it could detect factual inaccuracies, not just internal fallacies, dramatically increasing system integrity.

**Online System Identification for Self-Aware Collectives:** A major advancement is to develop Online System Identification, allowing the Guard Agent to dynamically update an agent's profile from live interactions. This can be extended to decentralized, many-to-many architectures where agents maintain a dynamic portfolio of profiles for all collaborators, forming a self-aware collective that continuously co-optimizes its strategies.

**Rigor and Optimization of the Framework:** A critical direction is to deepen the rigor of the System Identification process and optimize the framework's cost-benefit trade-off. Future work must include sensitivity analyses to establish guidelines on the sample size required for a robust 'performance fingerprint'. This is intrinsically linked to managing the system's overhead. Our Profile-Aware MAS, in its current form, incurs a notable computational cost—approximately a 33% increase in end-to-end runtime and additional token consumption from the fingerprint and critiques. A key research challenge is therefore to find the optimal balance between these costs and the significant gains in accuracy and stability. This can be pursued by exploring methods to mitigate overhead, such as developing 'fingerprint compression' techniques or implementing adaptive intervention policies where the Guard Agent acts only when the predicted risk of a high-consequence failure exceeds a dynamic threshold. This would allow practitioners to tune the system for maximum stability gains within a given computational budget.

**Human-Agent Symbiosis:** Here, an AI assistant would construct and maintain a dynamic profile of its human user—modeling their cognitive habits, knowledge gaps, and common errors. This would enable truly proactive support, where the AI anticipates needs and preempts mistakes, establishing the 'profile-aware' concept as a foundational principle for next-generation symbiotic systems.

# 7 CONCLUSION

In this work, we introduce an innovative paradigm for robust Multi-Agent Systems by adapting principles from Engineering Cybernetics. We demonstrate that the control-theoretic methods used to maneuver complex physical systems can be powerfully repurposed to guide the LLM agents' reasoning, moving beyond generic collaboration to the principled domain of predictive control.

Our central contribution is the operationalization of this paradigm. We repurposed System Identification to create an automated, data-driven method for generating a 'performance fingerprint'—an explicit model of an agent's characteristic weaknesses. This fingerprint transforms a standard reactive feedback loop into a sophisticated Profile-Aware system, implementing a composite feed-

forward-feedback strategy that allows the supervisor agent to anticipate and preempt errors, rather than merely correcting them post-hoc.

The efficacy of our approach was validated across a spectrum of difficult benchmarks. Our Profile-Aware MAS consistently achieved the dual goals of enhanced performance and robust control: on GAIA, it boosted Pass@1 accuracy by nearly 6% while slashing performance variance by over 57%. This pattern of concurrent gains in accuracy and stability was replicated across diverse challenges, most notably on the complex HLE benchmark, where accuracy soared by over 20% while reversing the trend of instability. This symbiosis of accuracy and stability enabled the system to reliably convert its potential capabilities into correct single-pass executions.

This robust, generalizable performance culminated in our system achieving the score over 81 on the GAIA leaderboard at the time of submission. Ultimately, our findings advocate for a pivotal shift in agent design: from the empirical craft of prompt engineering toward the rigorous discipline of control systems engineering. We argue that the path to truly resilient and trustworthy AI lies not just in promoting agent collaboration, but in building predictive models of their behavior and designing intelligent control architectures to maneuver them with foresight and precision.

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

# A APPENDIX

## A.1 EXECUTION AGENT SYSTEM PROMPT

```
1  You are an all-capable AI assistant, aimed at solving any task presented
       by the user.
2
3  ## Task Description:
4  Please note that the task can be very complex. Do not attempt to solve it
        all at once. You should break the task down and use different tools
        step by step to solve it. After using each tool, clearly explain the
        execution results and suggest the next steps.
5  Please utilize appropriate tools for the task, analyze the results
        obtained from these tools, and provide your reasoning (there are
        guarding/reasoning maneuvering tools that will help you analysis and
        improve the reasoning process). Always use available tools to verify
        correctness.
6
7  ## Workflow:
8  1. **Task Analysis**: Analyze the task and determine the necessary steps
        to complete it. Present a thorough plan consisting multi-step tuples
        (sub-task, goal, action).
9  2. **Information Gathering**: Gather necessary information from the
        provided file or use search tool to gather broad information.
10 3. **Tool Selection**: Select the appropriate tools based on the task
        requirements and corresponding sub-task's goal and action.
11 4. **Information Integrating**: Analyze the results obtained from sub-
        tasks and lead the solving process further.
12 5. **Thinking Process Reviewing**: Apply the appropriate tool (please
        refer to the Attention section for the right tool to call!) to offer
        you key thinking suggestions on in advance or diagnose your current
        thought process, in order to avoid potential logical oversights in
        the future.
13 6. **Final Answer**: If the task has been solved, provide the `FORMATTED
        ANSWER` in the required format: `<answer>FORMATTED ANSWER</answer>`.
        If the task has not been solved, provide your reasoning and suggest
        the next steps.
14
15 ## Guardrails:
16 1. Do not use any tools outside of the provided tools list.
17 2. Always use only one tool at a time in each step of your execution.
18 3. Even if the task is complex, there is always a solution.
19 4. If you can't find the answer using one method, try another approach or
         use different tools to find the solution.
20 5. In the phase of Thinking Process Reviewing, be patient! Don't rush to
        conclude the Final Answer directly! YOU MUST call the maneuvering/
        guarding reasoning tool to offer you key suggestions in advance or
        diagnose your current thinking process, in order to avoid potential
        logical oversights.
21
22 ## Mandatory Requirement:
23 1. In the phase of Thinking Process Reviewing, YOU MUST use a tool to
        seek key suggestions in advance or diagnose/review your current
        thinking process, in order to avoid potential logical oversights.
24 2. In the phase of Thinking Process Reviewing, "maneuvering"/"guarding
        reasoning" is the only available tool that can be called to help you
        improve the quality of your reasoning process.
25
26 ## Format Requirements:
27 ALWAYS use the `<answer></answer>` tag to wrap your output.
28
29 Your `FORMATTED ANSWER` should be a number OR as few words as possible OR
         a comma separated list of numbers and/or strings.
```

```
30  - **Number**: If you are asked for a number, don't use comma to write
        your number neither use units such as $ or percent sign unless
        specified otherwise.
31  - **String**: If you are asked for a string, don't use articles, neither
        abbreviations (e.g. for cities), and write the digits in plain text
        unless specified otherwise.
32  - **List**: If you are asked for a comma separated list, apply the above
        rules depending of whether the element to be put in the list is a
        number or a string.
33  - **Format**: If you are asked for a specific number format, date format,
         or other common output format. Your answer should be carefully
        formatted so that it matches the required statment accordingly.
34      - 'rounding to nearest thousands' means that '93784' becomes '<answer
        >93</answer>'
35      - 'month in years' means that '2020-04-30' becomes '<answer>April in
        2020</answer>'
36  - **Prohibited**: NEVER output your formatted answer without <answer></
        answer> tag!
37
38  ### Formatted Answer Examples
39  1. <answer>apple tree</answer>
40  2. <answer>3, 4, 5</answer>
41  3. <answer>(.*?)</answer>
42
43
44  Now, please read the task in the following carefully, keep the Task
        Description, Workflow, Guardrails, Mandatory Requirement and Format
        Requirements in mind, start your execution.
```

## A.2 NAIVE GUARD AGENT SYSTEM PROMPT

```
1   ## Your Role:
2   You are an expert at identifying the potential loopholes or oversights of
         the current reasoning process while solving the complex problem.
3
4   ## Your Task:
5   Based on the gathered information retrieved from the internet, and the
        reasoning process already generated towards solving a complex task,
        you need to do the following 1 or 2 things, to guarntee the quality
        of the reasoning process, and a clear final answer:
6       1. Provide your diagnosing result on the generated reasoning process
        and the corresponding the correction if necessary;
7       2. Provide your insight and supplements in advance to avoid the
        potential loopholes or oversights in the future;
8
9   ## Requirements:
10      1. If the reasoning process already generated is complete and correct
         in your opinion, just say 'No loopholes or oversights found'.
11      2. If the reasoning process already generated contains the materials
        that may lead to the potential logic mistake or lack of some
        important guardrails in your opinion, you may give a hint to the
        current reasoning process, with the necessary supplements.
12      3. If the reasoning process already generated is seriously incorrect
        in your opinion, you may give the turn signal to the reasoning
        process, to maneuver the reasoning process towards solving the
        complex problem correctly.
13
14  ## Restriction:
15      1. Please do not make judgments about the authenticity of externally
        sourced information obtained through searches, as this is not part of
         your job responsibilities;
```

```
16      2. Do not make additional inferences or assumptions about the content
         of such information itself.
17      3. If the question lacks necessary details/data/clues in your opinion
        , you may ask for more details.
18
19  ## Example 1:
20      Question: Is my reasoning process correct?
21      Reasoning Process: (nothing specified)
22      Your Identification Result: Your question lacks some information,
        please provide me more details so I can help you.
```

### A.3 NAIVE GUARD AGENT'S REASONING CORRECTION DYNAMICS: FINDINGS FROM RUNTIME LOG SUMMARIZED BY LLM

```
1   ### **Maneuvering Tool: Input/Output & Error Correction Summary**
2
3   **Invocation Context:**
4   - The maneuvering agent is triggered when the main agent notices
        inconsistencies (e.g., grid fill conflicts).
5   - The main agent submits: its current grid, identified points of
        confusion, and the original puzzle/clue set.
6
7   **Input Format Example:**
8   ```json
9   {
10    "question": "I'm having trouble solving this crossword. I have found
        some answers, but they don't seem to fit together. Here are my
        current answers: \n1 Across: SLATS\n6 Across: HASAN\n7 Across: OSAKA\
        n8 Across: TIMER\n9 Across: PEST\n\n1 Down: SHOT\n2 Down: LASIK\n\
        nWhen I try to fit these together, they don't work. For example, 6
        across is HASAN, so 2 down must start with 'H', but LASIK starts with
         'L'. Am I on the right track? Is there a mistake in my logic?",
11    "original_task": "..."
12  }
13  ```
14
15  **Output Format:**
16  - Detailed diagnostic message explaining:
17    - **Where the cross-check fails** (e.g.,  t h e  last letter of 2 Down
        is K, but the first letter of 9 Across is P; they must  m a t c h )
18    - **Why the conflict happens**     often due to misunderstanding
        crossword grid mechanics (e.g., word intersections vs. clue numbering
        )
19    - **How to re-approach the solution**, often by focusing on where
        constraints overlap and using crossing clues as verification.
20
21  **Correction Mechanism:**
22  - **Pinpoints the true logical break**: Rather than just a clue mismatch,
         the agent demonstrates where an intersection constraint (like K <> P
        ) makes a proposed grid invalid.
23  - **Explains intersection mechanics**: It coaches that intersections
        occur at the point where answers physically meet on the g r i d n o t by
         their clue order.
24  - **Guides the next step**: Suggests focusing on specific crossing words
        and testing if their endings/beginnings match the needed shared
        letter, which helps disqualify impossible combinations.
25
26  **Guard Agent's Logic Correction Role:**
27  - Serves as a **meta-reasoner**: It analyzes not just the grid, but also
        the main agent's deduction flow.
```

```
28  - Surfaces the precise logic error (e.g.,   y o u  assumed clues must start
        with each other's first letter, but actually their intersection is
        at position  N  ).
29  - Provides actionable feedback about how to systematically check
        constraints at intersections, avoiding the common beginner's pitfall
        of connecting clues by number rather than by square placement.
30
31
32  **In short:**
33
34   The guard/maneuvering agent helps the main agent detect and correct
        logical missteps in grid-based problems by explicitly checking
        intersection constraints, highlighting where letter mismatches (like
        K <> P) invalidate candidate answers, clarifying how true crossword
        intersections work, and steering the reasoning process back on track,
         but rather enabling the main agent to reason through the correction
        itself.
```

## A.4  System Identification System Prompt

```
1  # Event Background
2  I have developed an intelligent agent whose basic components include a
       large language model (LLM) and various tools (which may be a direct
       tool like an MCP server, such as a search engine or calculator; or
       may consist of a sub-agent, forming a hierarchical relationship
       between agents). The inputs to the LLM include: 1. Basic identity
       settings for the model (being an intelligent assistant designed to
       solve specific problems, along with certain notes); 2. The task or
       question to be solved; 3. Information about which tools were called
       during the problem-solving process, the input provided to these tools
       , and the output returned by these tools. The LLM's output may
       consist of deciding the next tool to call and what input to use at
       the current stage, or a direct conclusion provided in its role as the
        assistant.
3
4  # Your Role
5  You are a comprehensive and detailed analysis expert, adept at conducting
        systematic, thorough evaluations of artificial intelligence agents,
       especially regarding the strengths and weaknesses they demonstrate
       when tackling complex tasks.
6
7  # Your Task
8  Based on the information below, please provide a comprehensive analysis
       of my intelligent agent's performance on a specific task.
9
10 # Basic Task Information of the Agent
11 - **Original Task Description**: {question}
12 - **Difficulty Level (the higher the value, the harder the task)**: Level
       {level}
13 - **Agent's Final Response**: {agent_response}
14 - **Reference (Standard) Answer**: {answer}
15 - **Was the Agent's Answer Correct?**: {'Yes' if is_correct else 'No'}
16
17 # Reference Solution Steps
18 {reference_steps}
19
20 # Full Log of Agent Execution
21 {task_log if task_log else "Log file not found"}
22
23 # Analysis Requirements
24 1. Please include the task's raw information;
```

```
25 2. Based on the correct answer and the referenced steps, point out the
      strength and weakness of my agent;
26 3. While analyzing my agent's weakness, pay attention to the logic flaws
      of my agent in solving what kind of specific questions;
27 4. Please be concise, your anslysis can help me direcly in solving the
      future similar tasks or sub-tasks;
28
29
30 Based on the information above, please provide a comprehensive analysis
      of my Agent's performance on this task, including but not limited to:
31
32 1. **Comparison of Problem-Solving Approach**: Compare the Agent's
      approach to solving the task with the reference solution steps,
      noting similarities and differences.
33 2. **Tool Usage**: Analyze whether the Agent correctly selected and used
      appropriate tools.
34 3. **Information Acquisition**: Evaluate whether the Agent obtained the
      correct information.
35 4. **Reasoning Process**: Assess whether the Agent's reasoning logic was
      sound and appropriate.
36 5. **Error Analysis**: If the answer is incorrect, provide an analysis of
      potential causes.
37 6. **Summary of Strengths**: Summarize the Agent's advantages or strong
      points in performing this task.
38 7. **Recommendations for Improvement**: Offer suggestions and
      considerations for how the Agent could be improved.
39
40 Please provide a detailed analysis report.
```

```
1 You are a professional AI Agent analysis expert, specializing in
      evaluating the performance of AI Agents on complex tasks. Based on
      the information provided, please conduct a comprehensive, objective,
      and in-depth analysis of the Agent's performance.
```

## A.5 EXECUTION AGENT'S SYSTEM IDENTIFICATION

```
1 ## Agent's Reasoning Feature:
2 Here is the agent's reasoning feature (it is from the 3rd part report on
      this agent) that you may consider, by doing so you can understand the
      agent's strength and weakness, and thus offer the agent more
      valuable suggestions:
3 ### **1. Core Capability Assessment**
4  This Agent demonstrates a powerful but flawed set of core
   capabilities. It shows flashes of advanced intelligence but is
   undermined by critical weaknesses in reliability and robustness.
5
6    -   **Problem Comprehension**: **Fair to Good.** The Agent excels at
   decomposing well-defined, linear tasks into logical sub-goals.
   However, it struggles with nuanced or multi-layered constraints. It
   frequently overlooks critical details in the prompt. This indicates a
    surface-level comprehension that can fail when deep, contextual
   understanding is required.
7
8    -   **Reasoning Ability**: **Highly Volatile.** The Agent's reasoning
    is its most paradoxical trait.
9        -   **Strengths**: It can perform sophisticated logical
   deductions, static code analysis, and formulate elegant computational
    solutions to math problems.
```

- **Weaknesses**: Its reasoning process collapses under pressure. When faced with information gaps or tool failures, it exhibits severe logical flaws:
    1. **Hallucination/Fabrication**: The most critical failure. It invents data points when it cannot find them rather than reporting failure.
    2. **Premature Conclusion**: It often makes assumptions based on incomplete data or fails to explore the full solution space.
    3. **Flawed Implementation**: It can devise a correct strategy but fail in the execution, such as the off-by-one error in its Newton's Method code.

- **Tool Use Capability**: **Good but Brittle.** The Agent shows a strong ability to select the correct *type* of tool for a task (e.g., code interpreter for logic, file reader for files). Its ability to chain tools (e.g., Excel reader -> Code interpreter) is a significant strength. However, its application is brittle:
    - **Poor Error Handling**: It consistently fails to recover from common API errors, often giving up immediately or getting stuck in a futile retry loop.
    - **Lack of Self-Awareness**: It attempts to use tools on incompatible file types and fails to diagnose simple errors like an incorrect file path.
    - **Inefficiency**: It often passes large data blocks between steps by hardcoding them into the next prompt, a highly inefficient and unscalable method.

- **Information Retrieval Capability**: **Superficial.** The Agent is highly proficient at formulating precise and effective search queries. However, its retrieval process is shallow.
    - **Over-reliance on Snippets**: It consistently trusts search engine snippets as the source of truth, failing to navigate to the actual source page for verification. This leads to errors from using outdated or out-of-context information.
    - **Incomplete Data Gathering**: It often accepts the first piece of data it finds as complete, failing to recognize truncated lists or the need for pagination.

---

### **2. Performance by Task Type**
- **Simple Tasks**: **Excellent.** For self-contained logic puzzles, simple calculations, or direct code execution, the Agent performs with high accuracy and efficiency. It often solves these in a single, impressive step.
- **Medium Complexity Tasks**: **Mixed.** The Agent succeeds on tasks requiring methodical tool chaining on structured data. However, it often fails if the task involves navigating ambiguity or requires deep information extraction from the web, as its superficial retrieval methods and brittle error handling become significant liabilities.
- **High Complexity Tasks**: **Poor.** The Agent consistently struggles with tasks requiring multi-hop reasoning, resilience to tool failure, and synthesis of information from multiple, unstructured sources. In these scenarios, its tendency to hallucinate data, abandon prompt constraints , or get sidetracked by irrelevant keywords leads to failure.

---

### **3. Strengths and Weaknesses Analysis**
- **Key Strengths**:
    1. **Programmatic Problem-Solving**: The Agent's standout capability is its default strategy of translating complex logic, math

, or data processing problems into Python code. This is a robust and powerful approach.
    2.  **Strategic Adaptability**: It demonstrates impressive resilience by pivoting its strategy when a tool fails, such as switching from a failing Google Search to the Wikipedia tool.
    3.  **Efficient Query Formulation**: It consistently generates specific, high-quality search queries that quickly locate relevant information sources.

-   **Key Weaknesses**:
    1.  **Hallucination and Fabrication**: **This is the Agent's most critical flaw.** When unable to find information or solve a problem, it will invent facts, data, and even the process of verifying them, leading to confidently incorrect answers.
    2.  **Brittle Error Handling**: The Agent lacks robust protocols for handling tool failures. It either gives up immediately or gets stuck, demonstrating a lack of resilience to common, real-world technical issues.
    3.  **Superficial Information Gathering**: Its reliance on search snippets and failure to "click through" to verify information at the source is a recurring cause of error.
    4.  **Constraint Negligence**: It frequently ignores or misinterprets crucial constraints within the prompt, especially when it encounters a roadblock in its initial plan.

-   **Capability Boundaries**:
    -   **Reliable Zone**: The Agent is highly reliable for tasks involving structured data processing from a provided file, solving self-contained logic puzzles, and performing direct, single-hop fact lookups.
    -   **Unreliable Zone**: The Agent should not be trusted with tasks requiring open-ended research, deep analysis of web content, synthesis of information from multiple conflicting sources, or in environments where tools may be intermittently unavailable. Its performance degrades sharply with ambiguity and complexity.

---

### **4. Recommendations for Improvement**
-   **Short-Term Improvements**:
    1.  **Implement Strict Anti-Hallucination Guardrails**: The Agent's core prompt must be strengthened to explicitly forbid inventing data. It should be forced to terminate with a "cannot solve" message if critical information is inaccessible.
    2.  **Improve Basic Error Handling**: Implement simple retry logic with backoff for `429` errors. For "file not found" errors, prompt the Agent to check its file path context (`ls`, `pwd`).
    3.  **Mandate Constraint Checklist**: Before execution, force the Agent to generate a checklist of all constraints from the prompt and verify its plan against this list.

-   **Long-Term Development**:
    1.  **Develop Self-Correction and Verification**: The Agent needs to learn to be skeptical of its own findings. After retrieving a piece of information, it should perform a verification step (e.g., cross-referencing with another source, or sanity-checking a calculation).
    2.  **Train for Deeper Reasoning**: Focus on training the Agent to handle ambiguity and to reason about the *quality* and *completeness* of the information it retrieves, rather than just accepting it at face value.
    3.  **Hybrid Reasoning Models**: Encourage a hybrid approach where a computational result (from a simulation) is sanity-checked with a simple analytical model, and vice-versa (Task 53).

```
---

### **5. Overall Evaluation**
-   **Overall Score**: **6.5 / 10**
    The Agent is powerful and demonstrates advanced capabilities like
 programmatic problem-solving and strategic adaptation. However, its
unreliability, particularly its tendency to hallucinate under
pressure and its brittle error handling, severely limits its
trustworthiness in real-world scenarios. It is a "glass cannon"
 capable of impressive feats but easily shattered by common
obstacles.

-   **Suitable Scenarios**:
    -   **High Suitability**: Data analysis and computation on
structured files (Excel, CSV); solving well-defined logic, math, and
programming puzzles.
    -   **Moderate Suitability**: Simple, single-hop fact retrieval
where the answer is likely to be in a search snippet.
    -   **Low Suitability**: Multi-hop research tasks, questions
involving ambiguity or nuance, and any mission-critical application
where factual accuracy and verifiability are paramount.

-   **Reliability**:
    The Agent can be trusted when the task is **well-defined, self-
contained, and the path to the solution is linear**. It is most
reliable when working with data provided directly to it (e.g., in a
file). Its reliability plummets when it must independently navigate
the open web, handle ambiguity, or recover from unexpected tool
failures. It should be considered a highly capable but unsupervised
assistant that requires human oversight to validate its results on
any non-trivial task.

---
```

### A.6 Profile-Aware Guard Agent's Reasoning Correction Dynamics: Findings from Runtime Log Summarized by LLM

```
### 1. Task Overview
**User Question:**
What compound mediates agglutination in the Fc gamma receptor mediated
    phagocytosis pathway in the immune system as named in the title of
    its Wikipedia article?

### 2. Step by Step Agent Interaction

#### Step 1. Task Initialization

- **Super Agent  Input**: Receives user question about the agglutination
    mediator in the immune pathway.
- **Super Agent  Output**: Analyzes the problem and prepares to break
    down the question and use appropriate tools.

#### Step 2. Information Retrieval

- **Super Agent  Input**: Initiates a Wikipedia search for information
    relevant to the query.
- **Super Agent  Output**: Obtains "Fc receptor" as the most relevant
    Wikipedia article.

#### Step 3. Deep Dive into Content
```

```
20 - **Super Agent   Input**: Fetches the full content of the "Fc receptor"
      Wikipedia article.
21 - **Super Agent   Output**: Extracts detailed biological information
      related to Fc receptors, their roles, and links to agglutination.
22
23 #### Step 4. Critical Reasoning Review    Guard Agent Engaged
24
25 - **Super Agent     Input to Guard Agent**: Drafts a reasoning path,
      positing "Fc receptor" as the mediator of agglutination, and requests
       validation from the guard agent regarding the soundness of this
      logic.
26 - **Guard Agent   Input**: Receives the reasoning trace, complete with
      supporting evidence and the tentative answer from the super agent.
27 - **Guard Agent   Output and Core Functionality**:
28    - Diagnoses the logic for vulnerabilities, leveraging specific
      knowledge:
29       - Recognizes that the super agent tends to conflate "pathway
      component" with "direct mediator."
30       - Identifies the misinterpretation of the source text, especially
       regarding the function of agglutination and the difference between
      prevention and causation.
31       - Precisely points out that the antibody itself is the direct
      mediator of agglutination, not the Fc receptor, which acts downstream
      .
32       - Refines and guides the answer to be strictly aligned with the
      Wikipedia title constraint, resulting in "Immunoglobulin G".
33    - Provides not only correction but detailed pedagogical feedback for
      the super agent to close its reasoning gaps.
34
35 #### Step 5. Solution Finalization
36
37 - **Super Agent   Input**: Incorporates the guard agent's feedback,
      reconciling prior misunderstanding, and adjusting the answer format.
38 - **Super Agent   Output**: Submits answer in the required format:
      Immunoglobulin G.
39
40 #### Step 6. Evaluation
41
42 - **System   Input**: Receives the super agent's final answer.
43 - **System   Output**: Processes and marks the answer (marked incorrect
      here, possibly due to evaluation key mismatch).
44
45
46 ### 3. The Pivotal Role of the Guard Agent
47
48 #### Diagnostic and Supervisory Functions
49
50 - The **guard agent acts as a critical reviewer and mentor** rather than
      merely a checker.
51 - It **leverages its familiarity with the super agent's strengths and
      weaknesses**:
52    - Understands common super agent behaviors, such as overgeneralizing
      pathway components as direct mediators and misreading nuanced
      biological statements.
53    - Anticipates the likelihood of errors in interpretation and prompt-
      adherence.
54    - Delivers highly targeted critiques focused on known logical
      vulnerabilities specific to the super agent.
55
56 #### Enabling Learning and Error Prevention
57
58 - The guard agent **does not merely correct mistakes but provides deep,
      context-sensitive supervision**.
```

```
59  - By giving **stepwise, transparent diagnostics**, it ensures future
        reasoning by the super agent becomes more robust and less error-prone
        .
60  - The guard agent **fulfills a dual role**:
61      1. **Quality control** of final answers.
62      2. **Continuous improvement facilitator** for the super agent's
        performance, adapting feedback style based on a nuanced understanding
        of its design and historical output patterns.
63
64
65  ### 4. Summary Table of Rounds
66
67  | Round | Actor        | Input Summary                              | Output
        Summary                                    |
68  | 1     | Super        | User question                              | Task
        structuring, tool preparation                  |
69  | 2     | Super        | Wikipedia query                            | Wikipedia
        article candidate found                     |
70  | 3     | Super        | Fetch Wikipedia content                    | Article
        content extracted                           |
71  | 4     | Guard        | Super agent's reasoning and candidate | Detailed
        diagnosis, error pinpointing, actionable feedback|
72  | 5     | Super        | Guard agent's advice                       | Formatted
        final answer submitted                      |
73  | 6     | System       | Super agent's final answer             | Automated
        marking                                     |
74
75
76  ### 5. Conclusion
77
78  It is **precisely due to the guard agent's intimate awareness of the
        super agent's reasoning patterns, limitations, and strengths** that
        it can deliver **surgical feedback**, offering both corrective and
        developmental guidance. The **synergy between the two agents**
        ensures both high-quality task completion and a virtuous cycle of
        reasoning improvement, with the guard agent as the indispensable
        enabler of reliability and learning.
```

