# OpenReview forum: "Profile-Aware Maneuvering: A Dynamic Multi-Agent System to Robust Agentic Problem Solving"
_ICLR.cc/2026/Conference — Submitted to ICLR 2026_

### Official Review · Reviewer_3mCR · 2025-10-30

**Soundness:** 2
**Presentation:** 2
**Contribution:** 2
**Rating:** 4
**Confidence:** 3

**Summary:**

This paper proposes a control theoretic framework for building robust multi agent systems. The authors model the execution agent as the plant and the guard agent as the controller. They begin with a reactive feedback design and then introduce a profile aware architecture that adds a predictive feed forward channel. The predictive channel is driven by a performance fingerprint learned offline through a system identification procedure motivated through the “zig-zag” test in marine vessels. Across the GAIA, HLE, and GPQA Diamond benchmarks, the profile aware system raises Pass at one accuracy and reduces variance. The paper reports gains in accuracy, stability, and gap reduction between Pass at three and Pass at one.

**Strengths:**

- Clear conceptual bridge between engineering cybernetics and agent design. The plant and controller mapping is precise, and the paper grounds the argument with a formal disturbance rejection analysis that motivates the feed forward path.
- The performance fingerprint is an interpretable policy that equips the guard agent to anticipate characteristic errors of its partner rather than only reacting after the fact. This addresses a frequent weakness in agent supervision

**Weaknesses:**

##

- The offline system identification stage is an important component for the Guard Agent, yet the description of data selection, sampling strategy (apart from random), and sensitivity to the number of logs is not fully specified. A reader will want concrete guidance about how large a sample size is required to obtain a reliable fingerprint perhaps depending on the task complexity.
- The evaluation controls model temperature and repeats invalid outputs, but it is unclear how the guard agent’s interventions affect context growth and tool latency. A cost and latency analysis would help practitioners adopt the method at scale.
- Presentation issues:
    - I find it hard to understand the effectiveness of the Laplacian analogy terms to the LLM agent terms, namely the existence of an ideal feedforward controller in the Laplacian space does not mean the Guard agent learns such a function.  The paper would benefit from bounds that reflect the mismatch between the learned fingerprint and the true plant, along with conditions under which feed forward control can degrade performance.
    - Minor: The quotation marks are not typeset correctly. In LaTeX, please use `` ` ``  (a grave accent) for opening single quotes and `'` (an apostrophe) for closing single quotes. e.g. L194 ‘G’

**Questions:**

1. How large is the context budget cost of injecting the fingerprint and the guard critiques, and what is the impact on throughput and latency in realistic settings?
2. Are the SAS and MAS settings given an equal computation/prompting budget? How are they considered equal footing?

---

> ### Author Response · Authors · 2025-11-20
>
> We are very grateful to the reviewer for their insightful, precise, and highly constructive feedback. These comments help us see clearly how to strengthen the paper's rigor and practical relevance.
>
> 1. On the Gap Between Ideal and Learned Control: This is a deep and perceptive question. We completely agree that our Guard Agent is an engineering approximation of the ideal feed-forward controller, not a perfect mathematical realization of C_ff (s)=−1/P(s).
>
> The formal analysis in Section 2.3.4 serves to establish the theoretical motivation—it proves why a feed-forward architecture is desirable. Our contribution is an engineering methodology (System ID via log analysis) to construct a practical system that strives toward this theoretical ideal.
>
> The reviewer's point about the mismatch is crucial. Our performance fingerprint is our method for creating an empirical model of the plant's predictable behaviors (P(s)) and, more importantly, its characteristic disturbances (D(s)). An inaccurate fingerprint could indeed degrade performance, which underscores why having a principled, offline System Identification stage is so critical and why our method is distinct from simple online reflection. We will add a paragraph to the manuscript that explicitly discusses this "model mismatch" and how the quality of the offline identification directly impacts the efficacy of the online feed-forward control.
>
> 2. On System Identification Details: This is an excellent point. Our description was brief. We acknowledge that a sensitivity analysis on the required sample size for generating a reliable fingerprint is a critical area for future work. Intuitively, this size likely depends on task complexity and the base agent's error frequency. We will add this to our Future Work section.
>
> 3. On Cost and Latency Analysis: Thank you for highlighting this. Our method introduces a trade-off. We have measured that our Profile-Aware MAS increases end-to-end runtime by approximately 33% against the SAS baseline. It also adds a fixed token cost for the fingerprint and a variable cost for each critique. We will add a new subsection to formally discuss this cost-benefit analysis, framing our performance gains in the context of this additional computational budget.
>
> 4. On Budget Equality: This is a key aspect of our experimental design. The SAS and MAS settings are intentionally not given an equal computation budget. Our scientific goal is to measure the return on investment for allocating additional computational resources to a supervisory controller. Our results demonstrate that this added budget, when structured within our predictive control framework, yields disproportionate gains in both accuracy and, most critically, system stability (variance reduction). We will clarify this experimental philosophy in the paper.
>
> Thank you again for helping us sharpen our arguments and improve the paper with your expert feedback.

---

### Official Review · Reviewer_3avE · 2025-11-01

**Soundness:** 2
**Presentation:** 2
**Contribution:** 2
**Rating:** 2
**Confidence:** 4

**Summary:**

This paper approaches LLM agents from the perspective of control theory. The idea is to capture the “system’s unique behavioral fingerprint” and then design a “control architecture” to engineer “reliable LLM-based agents.” The approach is evaluated on subsets of GAIA, HLE, and GPQA Diamond.

**Strengths:**

Using principles from control theory for agent control is an interesting idea that I haven’t seen before, at least not so directly/explicitly as in this paper.

**Weaknesses:**

* While the overall motivation is interesting, in practice, the implementation of the framework amounts to using LLMs as judges for “System identification” and building the “Overall profile.” I don’t see anything substantially different from many other frameworks such as ReAct that involve some kind of reflection. The underlying stochastic nature of many of these steps makes the application of control theory a lot more fraught.
* If there are in fact substantive differences from previous LLM-as-judge approaches, these need to be made more explicit in the framing of the paper.
* These concerns may be mitigated by including systematic comparisons to baseline systems and showing empirical outperformance. However, there do not appear any meaningful baseline comparisons in the paper.
* Critical experimental details, such as how model selection was performed (budget, validation dataset, etc.), are not included.
* Figure 1 illustrates a setting that is so far removed from the setting under consideration that it’s a questionable use of space.
* There is insufficient discussion of the limitations of the approach. For example, the approach appears to incur significant computational expense, both online and offline.
* Large portions of the text appear to be written by an LLM.

**Questions:**

> Our experiments utilize 109 questions (the specific task IDs will be released later on GitHub

> Humanity’s Last Exam (HLE) is a challenging benchmark from the Center for AI Safety and Scale
AI, featuring 2,500 questions across a wide range of subjects. We randomly selected a subset of 100
questions (47 Math, 15 Biology/Medicine, 11 Computer Science/AI, 10 Physics, and 17 from other
fields, task IDs will be released later)

Why not use a standard evaluation split and compare to baselines?

---

> ### Author Response · Authors · 2025-11-20
>
> We thank the reviewer for their thoughtful critique and for recognizing the interesting motivation of our work. We would like to address the concerns raised, particularly regarding the distinction from other frameworks.
>
> 1. On Substantive Differences from ReAct and Reflection Frameworks: While our work involves an agent providing feedback, it is fundamentally different from ReAct or other reflection-based methods in its theoretical foundation, its operating philosophy, and its objective.
>
> Theoretical Grounding: Frameworks like ReAct, Reflexion, and Expel are inspired by cognitive science—they aim to mimic human reasoning patterns like "thinking before acting" or "learning from mistakes." Our work is grounded in Engineering Cybernetics. We do not aim to mimic cognition; we aim to engineer a stable and predictable system by applying the principles of control theory.
>
> Error Handling Philosophy (Predictive vs. Reactive): This is the core distinction. ReAct synergizes reasoning and acting in real-time. Reflexion is post-hoc, reflecting on a completed trajectory to improve the next one. Our Profile-Aware MAS is proactive and predictive. Thanks to the offline System Identification process, our Guard Agent possesses a model (performance fingerprint) of the Execution Agent's characteristic failures. To use an analogy: ReAct is a driver who thinks "I see a red light, I should brake." Reflexion is a driver who thinks "I ran that red light, next time I'll be more careful." Our Guard Agent is an expert driving instructor who knows the student tends to brake late, and proactively says, "This intersection is tricky, prepare to brake early." This feed-forward control, which preempts errors, is a capability that purely reactive or real-time systems lack.
>
> "Knowledge" Used: ReAct and Reflexion use knowledge about the task. Our system creates and uses meta-knowledge about the agent itself. This is a higher level of abstraction and control.
>
> 2. On Stochasticity and Control Theory: The reviewer correctly notes the stochasticity of LLMs. This is precisely why control theory is so applicable. Control systems are designed to govern systems rife with noise and disturbances. Our results, which show a dramatic and consistent reduction in performance variance (Tables 1 & 2), provide strong empirical evidence that our control architecture successfully dampens the negative effects of this inherent stochasticity, leading to a more reliable system.
>
> 3. On Baselines and Experimental Details: Our experimental design employs ablative analysis to scientifically isolate the contribution of each architectural component (tools, reactive control, predictive control). For data splits on HLE/GPQA, we simulated a standard train/validation procedure where the "System Identification" phase acts as the validation/tuning step. We will clarify this and release all task IDs for reproducibility.
>
> 4. On Computational Expense: This is a fair and important criticism. Our Profile-Aware MAS does incur an additional cost. Based on our experiments, it increases the total runtime by approximately 33% compared to the SAS baseline. This represents a clear trade-off: increased computational cost for significant gains in accuracy, reliability, and predictability. We will add a dedicated discussion on this limitation and cost-benefit analysis in our revision.

---

### Official Review · Reviewer_ijYn · 2025-11-01

**Soundness:** 2
**Presentation:** 1
**Contribution:** 2
**Rating:** 2
**Confidence:** 3

**Summary:**

The authors propose a Profile-Aware Maneuvering (PAM) approach, inspired by control theory and system identification in engineering. The approach involves reframing agent design as a control systems problem, focusing on predictive rather than reactive strategies. A Multi-Agent System (MAS) is developed, where a Guard Agent acts as a predictive controller to preemptively correct the Execution Agent’s errors using a performance fingerprint. Tested on benchmarks like GAIA, HLE, and GPQA Diamond, the Profile-Aware MAS demonstrated superior performance to the baseline single-agent LLM and vanilla MAS.

**Strengths:**

1. This work proposes a complete, well-defined control system for giving valuable feedback to LLM agent systems for controlling them towards desired directions.

**Weaknesses:**

1.The presentation can be improved. Given the complexity of the control system, it is unclear why this framework is suitable for, or how it can be projected onto, LLM systems—both at the intuitive level and in the mathematical formulation.

2. The inputs and outputs of the multi-agent LLM system are not clearly introduced. The notation is overly complicated and lacks a coherent logical flow, which makes it difficult for the reviewer to follow. For example, what is the input to the guard agent, and what are the inputs to the other agents under its guidance? Do they receive the same input or different inputs? Furthermore, at each time step, the mechanism by which the agents’ outputs are aggregated to inform the next step is not clearly explained. The reviewer recommends presenting a complete pipeline in a dedicated section or subsection. Additionally, Figure 2 should include corresponding notation that matches the text, which would make the overall process clearer.

**Questions:**

1. What does the "new tool information Info_t" refer to? Could the author provide some examples or a detailed explanation? (line 157)

---

> ### Author Response · Authors · 2025-11-20
>
> We thank the reviewer for their valuable feedback and for highlighting areas where the presentation could be significantly improved. We agree that a clearer explanation of the system's operational flow is crucial for understanding our contribution.
>
> 1. On Clarity and System Pipeline: We apologize that the logic flow and notation were difficult to follow. We will revise the manuscript to provide a more intuitive and step-by-step description. The core idea is to move from a reactive to a predictive control system.
>
> The Naive MAS (Reactive Feedback): The Execution Agent works on a task. When it gets stuck, it can call the Guard Agent. The Guard Agent then reviews the past actions and provides a critique to correct the error. This is a classic feedback loop, correcting errors post-hoc.
>
> The Profile-Aware MAS (Predictive Control): This is our main contribution. Before any online task begins, we perform an offline System Identification to create a performance fingerprint—an explicit summary of the Execution Agent's typical mistakes. During the online task, this fingerprint is injected into the Guard Agent's context. Now, when the Guard Agent reviews the Execution Agent's work, it doesn't just look for current logical fallacies; it also uses the fingerprint to proactively anticipate potential errors based on the Execution Agent's known weaknesses and provides preemptive guidance.
>
> 2. On Agent Inputs/Outputs: To clarify:
> Execution Agent Input: Task description + its own reasoning history including tool calls with arguments + tool outputs (Info_t) + any critiques from the Guard Agent.
>
> Guard Agent Input: The Execution Agent's specific question regarding the current reasoning + the original task info. You may refer this to Appendix 3. In the Profile-Aware MAS, its context is also permanently augmented with the performance fingerprint.
>
> The two agents interact sequentially. The Guard Agent's output (a critique) becomes a new input for the Execution Agent's next reasoning step, thus closing the control loop.
>
> 3. On Figure 2 and Info_t: These are excellent suggestions. We will revise Figure 2 to explicitly include the control theory notation (P, C, u, y, etc.) to make the architecture-to-theory mapping unambiguous. We will also clarify that Info_t refers to the output from any external tool called by the agent (e.g., the text from a web search, or the numerical result from a code interpreter), and will include examples in the text.
>
> We appreciate these constructive comments and are confident that addressing them will make our paper much stronger and more accessible.

---

### Official Review · Reviewer_sPdF · 2025-11-03

**Soundness:** 1
**Presentation:** 1
**Contribution:** 2
**Rating:** 0
**Confidence:** 3

**Summary:**

The authors aims to improve LLM agent orchestration by viewing the problem from the lens of control theory, proposing a “Guard Agent” that generates feedback by concatenating the original input with a collection of task traces (which they denote the “performance fingerprint”) collected offline. Experimental results on the GAIA benchmark show that including this “performance fingerprint” improves performance compared to without.

**Strengths:**

- To the best of my (very limited) knowledge in this domain, the framing of the model evaluation problem as “system identification” appears to be novel

**Weaknesses:**

- There is no related works section
    - Though I am not an expert in this field, it seems like there are many existing works that are very similar (e.g., [A, B])
- There are no comparisons to existing methods at all.
- The mapping to control theory is mostly qualitative. There is basically no connection between the math and the actual method
- The framing of the proposed "performance fingerprint" as feedforward planning is completely heuristic

[A] Shinn, Noah, et al. "Reflexion: Language agents with verbal reinforcement learning." *Advances in Neural Information Processing Systems* 36 (2023): 8634-8652.
[B] Zhao, Andrew, et al. "Expel: Llm agents are experiential learners." *Proceedings of the AAAI Conference on Artificial Intelligence*. Vol. 38. No. 17. 2024.

**Questions:**

- The connection to vessel maneuvering is very strange. There doesn’t seem to be anything specific to vessel maneuvering that isn’t also true for general systems in control.
- Some of the citations are very strange. Why cite system identification papers for ships instead of the dozens of classical system identification papers / reviews like [a]?
    - The citations for “vessel maneuvering” are also very strange. The paper is very recent and only has 11 citations even though there are classical papers / reviews such as [b].

[a] Åström, Karl Johan, and Peter Eykhoff. "System identification—a survey." *Automatica* 7.2 (1971): 123-162.

[b] Fossen, Thor I. *Guidance and Control of Ocean Vehicles*. John Wiley & Sons Limited, 1995.

---

> ### Author Response · Authors · 2025-11-20
>
> We thank the reviewer for their time. However, we must firmly state that the assessment of our work as "heuristic" and lacking in novelty stems from a fundamental misunderstanding of its core principles. We will address the points raised to clarify these misconceptions.
>
> 1. On Fundamental Novelty and Distinction from Reflexion/Expel: We respectfully but strongly disagree that our method is "very similar" to existing works like Reflexion or Expel. The comparison is misplaced as they operate on entirely different principles, timelines, and theoretical foundations.
>
>   Reflexion is a post-hoc, reactive framework. It generates verbal self-reflection after a task trajectory is completed, enabling iterative learning within a single task. Its learning is ephemeral and aimed at immediate self-correction.
>
>   Our Profile-Aware Maneuvering introduces a paradigm shift from cognitive mimicry to principled control engineering. Its core is an a priori modeling stage and proactive online control.
>
>   A Priori Modeling (Offline System Identification): We systematically analyze the agent's behavior on a separate dataset to generate a persistent, explicit policy—the performance fingerprint. This is not knowledge about the task (e.g., "how to find a file"), but meta-knowledge about the agent's own failure modes (e.g., "this agent tends to hallucinate Python libraries").
>
>   Proactive Control (Online Feed-Forward): This fingerprint allows the Guard Agent to act as a predictive, feed-forward controller, anticipating and preempting errors before they occur. This is fundamentally different from reacting to failures after the fact.
>
>   To use an analogy, Reflexion is like a person learning from a mistake after they've made it. Our method is like an aircraft's autopilot which uses a precise model of the plane's flight characteristics (our "fingerprint") to proactively counteract turbulence, ensuring a smooth flight.
>
> 2. On the Control Theory Mapping: The claim that our mapping is "heuristic" overlooks the formal mathematical justification in Section 2.3.4. The Laplace transform analysis is not a mere qualitative analogy; it is a formal proof demonstrating the theoretical superiority of a composite feed-forward-feedback architecture (our PA-MAS) for rejecting predictable disturbances. The performance fingerprint (P_E) is the practical implementation of the disturbance model (D(s)), and it is precisely this predictive capability that leads to the empirically validated, dramatic reduction in performance variance seen in our results (Tables 1 & 2).
>
> 3. On Related Work and Baselines: We will add a dedicated Related Work section to better contextualize our paper. However, our experimental setup already provides a rigorous ablative analysis (SAS vs. MAS vs. PA-MAS), which is a scientifically valid method for demonstrating the incremental value of each of our proposed architectural components. This isolates the source of performance gains far more effectively than comparing against a completely different framework would.
>
> 4. On Citations: The vessel maneuvering analogy is a deliberate pedagogical choice to make the abstract concept of System Identification tangible for the AI community. It is arguably the most intuitive real-world example of identifying a system's unique "fingerprint" to enable precise control. We cited recent work as is standard practice, but will add classical citations to acknowledge the field's deep history.
>
> Our work represents a deliberate shift from the "art" of prompt engineering towards the "science" of control systems engineering for building reliable agents. We believe this perspective is both novel and critical for the future of AI.

---

> > ### Comment · Reviewer_sPdF · 2025-11-25
> >
> > **Re: Related Works**
> >
> > - My concern is not with the novelty, but rather that it seems like existing works were not properly acknowledged and compared with.
> > - While the authors perform an ablation study, which helps readers understand the proposed framework, it is not possible to place the current work among the existing literature without a more thorough discussion and experimental comparison.
> > - The authors add a paragraph to the revision that now cites additional works. However, this paragraph does not describe the concrete differences between in methodology between the works, only using vague phrases such as “synergizes reasoning and acting in real-time”
> >
> > **Re: Mapping to Control Theory**
> >
> > - The author’s response seems to contradict what they write in the paper:
> >     - “The performance fingerprint (P_E) is the practical implementation of the disturbance model (D(s))”
> >     - Line 276: “The performance fingerprint $P_E$ serves as our empirically-derived model of the plant $P(s)$
> > - Whether the proposed “performance fingerprint” $P_E$ can be thought of as implementing either is questionable. There’s no evidence or experiments that justify this framing.
> > - The authors claim that the feedforward term $C_{ff}(s) = -1/P(s)$ is implemented by the Guard Agent
> >     - This claim is not well supported either. It could be that the effect of the Guard Agent is in increasing the “controller gain”, which would still improve the performance of the system
> >
> >
> >
> > **Re: System ID and Citations**
> >
> > - The claim of “arguably the most intuitive real-world example” is highly doubtful
> > - Citing recent work while ignoring influential classical works, or even other recent work that is more influential is not standard practice and is highly suspicious.

---

### Meta-Review · Area_Chair_Ho6o · 2026-01-11

**Summary:**

The paper receives virtually no post-rebuttal discussion from the reviewers except one follow up by the sPdF without follow-up response from the authors. The AC read the comments from the reviewers and authors and skim through the paper. The authors response is not completely covering reviewers' questions (eg, quesitons of sPdF, 3avE and 3mCR are not well responded by the authors). But the authors rather reply to them with highly decorative words (""core distinction"", ""crucial"", ""key aspect"", ""excellent point""). If the reviewer's ciritque is crucial, the paper has a serious flaw thus needs a significant revision before the acceptance. The AC concerns that the reviewers' concerns are not well addressed by the authors' response and needs significant revision before acceptance. Thus, the AC recommends the authors to revise and submit to a relevant venue.

**Reviewer Concerns:**

Most of the reviewers' concerns are not well addressed by the authors' response.

**Reviewer Scores:**

All reviewers except one have participated in the discussion. The only one reviewer discussing the comments were also not very active.

---

### Decision · Program_Chairs · 2026-01-26

Reject